# CTX-CNF1 Recombinant Protein Selectively Targets Glioma Cells In Vivo

**DOI:** 10.3390/toxins13030194

**Published:** 2021-03-08

**Authors:** Eleonora Vannini, Elisabetta Mori, Elena Tantillo, Gudula Schmidt, Matteo Caleo, Mario Costa

**Affiliations:** 1Neuroscience Institute, National Research Council (CNR), via G. Moruzzi 1, 56124 Pisa, Italy; elena.tantillo@gmail.it (E.T.); caleo@in.cnr.it (M.C.); costa@in.cnr.it (M.C.); 2Fondazione Umberto Veronesi, 20122 Milan, Italy; 3Scuola Normale Superiore, 56126 Pisa, Italy; elisabetta.mori@sns.it; 4Medizinische Fakultät, Institut für Experimentelle und Klinische Pharmakologie und Toxikologie, University of Freiburg, 79085 Freiburg, Germany; gudula.schmidt@pharmakol.uni-freiburg.de; 5Department of Biomedical Sciences, University of Padua, 35122 Padua, Italy

**Keywords:** glioblastoma, drug discovery, cytotoxic necrotizing factor type 1, chlorotoxin, recombinant protein production, glioma, brain tumor, drug delivery

## Abstract

Current strategies for glioma treatment are only partly effective because of the poor selectivity for tumoral cells. Hence, the necessity to identify novel approaches is urgent. Recent studies highlighted the effectiveness of the bacterial protein cytotoxic necrotizing factor 1 (CNF1) in reducing tumoral mass, increasing survival of glioma-bearing mice and protecting peritumoral neural tissue from dysfunction. However, native CNF1 needs to be delivered into the brain, because of its incapacity to cross the blood–brain barrier (BBB) per se, thus hampering its clinical translation. To allow a non-invasive administration of CNF1, we here developed a chimeric protein (CTX-CNF1) conjugating CNF1 with chlorotoxin (CTX), a peptide already employed in clinics due to its ability of passing the BBB and selectively binding glioma cells. After systemic administration, we found that CTX-CNF1 is able to target glioma cells and significantly prolong survival of glioma-bearing mice. Our data point out the potentiality of CTX-CNF1 as a novel effective tool to treat gliomas.

## 1. Introduction

Gliomas are the most common primary brain tumors, accounting for 30% of all primary brain tumors and 80% of all malignant ones. Glioblastoma (GB) is the most aggressive form, making up 54% of all brain tumors, affecting 3 in 100,000 people with nearly 23,000 cases per year worldwide. The current standard-of-care is represented by the surgical resection of the tumoral mass followed by cycles of radio- and chemotherapy. However, this standard protocol of intervention is not selective and the patients survival rate is about 12–15 months after diagnosis [1]. Hence, developing new approaches to counteract this terrible disease is necessary and represents one of the hardest challenges of our time for neuro-oncologists. 

In the last few years our group has studied the effects of a bacterial protein toxin called CNF1 (cytotoxic necrotizing factor 1) on GB. We have demonstrated that CNF1 triggers a long-lasting activation of intracellular Rho GTPases leading to multinucleation, senescence and eventually death of both murine and human glioma cells. When administered in vivo, CNF1 increases the survival of glioma-bearing mice and enhances neuronal function and plasticity, sparing neuronal responses in peritumoral areas [2,3,4]. Despite the promising results obtained in preclinical models, CNF1 translatability to clinics is limited by its inability of crossing the blood–brain barrier (BBB) in normal conditions [5]. Recent studies have proven that Chlorotoxin (CTX), a 36 amino acid peptide derived from the venom of the scorpion *Leiurus quinquestriatus,* is able to penetrate the BBB and to selectively recognize and target glioma cells [6,7,8,9]. Indeed, CTX-based bioconjugates are currently employed in clinics not only during the removal of the tumor mass, but also as drug vectors in clinical trials [10,11].

Here we developed a new chimeric protein called CTX-CNF1, created by the fusion of CTX with CNF1. This recombinant toxin is able to cross the BBB and to selectively target glioma cells, inducing the activation of a senescence process. Thanks to CTX peculiarities, this recombinant protein can be systemically administered and represents a novel, promising approach of high translational value for glioma treatment.

## 2. Results

### 2.1. CTX-CNF1 Affects Vitality of Both GL261 and PDGF+ TRP53−/− Glioma Cells

In order to find the effective half inhibitory dose (IC50) of CTX-CNF1 (Figure 1A), the chimeric protein was administered to GL261 cells at different concentrations (i.e., 1, 12, 25, 30, 40 and 50 nM). Regardless of the concentration, the vitality of GL261 cells was statistically affected after 48 h of CTX-CNF1 treatment (Figure 1B; One Way ANOVA *p* < 0.001). Because 25 nM was sufficient to reduce the cell vitality of almost 50%, this was the selected chimeric concentration used for the following in vitro assays. In this condition (i.e., 48 h of 25 nM CTX-CNF1 treatment) we also observed that 53.25% of cells appeared multinucleated (Figure 1B), in line with the well-known effect produced by naïve CNF1 [4,12,13,14,15]. With the betagalactosidase assay, performed at different time points after CTX-CNF1 administration, we found that more than 50% of GL261 cells were senescent after only 24 h of chimeric protein addition (Figure 1D; One Way ANOVA *p* < 0.001). We also confirmed the activation of a senescence process performing RT-PCR on treated and un-treated GL261 and U87 cells (Figure 1E,F). After 6 days of treatment, we found that CTX-CNF1 (25 nM) was able to induce a significant upregulation of senescence markers (i.e., p16 and p21; GL261, p16: T test, *p* = 0.0001; GL261, p21: T test, *p* = 0.0164; U87, p16: T test, *p* = 0.013). 

In order to better evaluate the CTX-CNF1 effect on gliomas we chose to test it also on another GB model, whose cells were GFP^+^. The selected cell line was the PDGF^+^ TRP53^−/−^ [16]. The number of PDGF^+^ TRP53^−/−^ total cells and spheroids were evaluated 72 h and 6 days after treatment with 25 nM CTX-CNF1. We found that the total number of cells significantly decreased after 6 days of treatment (one way ANOVA, *p* < 0.001; Figure 1G), but 72 h of chimeric administration was already sufficient to reduce the number of PDGF^+^ TRP53^−/−^ spheroids (one way ANOVA, *p* < 0.001; Figure 1H). Altogether, these analyses confirmed that CNF1 maintains part of its functionality [12,14,17,18] even when it is fused with CTX into a chimeric protein.

### 2.2. CTX-CNF1 Increases Survival of Glioma-Bearing Mice 

Twelve days after GL261 cells injection into the visual cortex (i.e., at the appearance of first symptoms of the disease; [2]), glioma-bearing mice were treated with CTX-CNF1 (*n* = 9) or vehicle solution (i.e., PBS; *n* = 9). A group of animals, that received a CTX-CNF1 (2 μL, 80 nM) injection into the lateral ventricles (Figure 2A), showed a significantly increase in survival rate (Figure 2B; *p* = 0.0018). 

In order to assess whether the chimeric protein could also produce an enhancement of survival when systemically given, we decided to deliver CTX-CNF1 (*n* = 6) or Vehicle solution (i.e., PBS, *n* = 9) through the caudal vein (20 μL, 80 nM; Figure 2C). It is worth noticing that this way of administration resembles the classic chemotherapy. Remarkably, CTX-CNF1 was effective in prolonging life expectancy even when intravenously administered (Figure 2D; *p* = 0.013), suggesting that CTX manages to cross the BBB and target glioma mass even when it is conjugated with CNF1. We also checked whether the two toxins alone, CNF1 and CTX, could produce an enhancement of glioma-bearing mice survival when systemically administered. We reported that CNF1 alone wasn’t effective in producing a significant increase in survival rate of glioma-bearing animals (*n* = 5; Figure 2F, *p* = 0.1). In particular, only two days after the treatment the systemic administration of CNF1 alone resulted to be lethal for the 66% of animals (Figure 2F). On the other hand, the intravenous administration of CTX alone produced a significant increase in glioma-bearing mice survival (*n* = 5; Figure 2H, *p* = 0.03). However, in order to understand whether the two treatments able to induce a significant augmented survival rate in glioma-bearing mice (i.e., CTX-CNF1 and CTX) could produce an effect also on glioma proliferation, we performed immunostaining for the Ki67 marker 24 and 48 h after systemic drug delivery. We found that only CTX-CNF1 intravenous administration produced a significant decrease of the proliferation rate, 48 h after delivery (Figure 2I; One Way ANOVA *p* = 0.001), whereas CTX alone wasn’t effective in acutely dampening glioma growth. 

Next, we also evaluated the possibility that CTX-CNF1 could recruit more microglial cells within the tumor microenvironment. In order to address this issue, we calculated the number of microglial (Iba1) and macrophages (Mac2) cells in the peritumoral tissue (100–300 μm from the tumor burden) of treated and un-treated mice. No differences between groups were detected for Iba1 (Figure 2L, *t* test *p* = 0.19), whereas CTX-CNF1 glioma-bearing animals showed an increase in peritumoral expression of Mac2 (Figure 2M, *t* test *p* < 0.001). 

### 2.3. CTX-CNF1 Specifically Targets Glioma Cells In Vivo 

The selective recognition of glioma cells by CTX-CNF1 was assessed through immunostaining (Figure 3A–D) and Western blot (Figure 3E) assays. To evaluate the time course that CTX-CNF1 required to target glioma cells, we performed immunostaining at two different time points after PDGF^+^ TRP53^−/−^ or GL261 cells inoculation. PDGF^+^ TRP53^−/−^ cells were easily identifiable because of GFP expression (Figure 3A–C). Hoechst was used to label every cell nuclei and, thanks to an antibody versus the CNF1 catalytic domain, it was possible to visualize the localization of the toxin. Remarkably, CNF1 was found to be adhered to PDGF^+^ TRP53^−/−^ cells membrane 24 h after its administration (Figure 3B,C). With confocal microscopy we also detected CTX-CNF1 in GL261 tumoral mass 6 days after its administration (Figure 3D). It is worth noticing that, at this time point, the chimeric protein was found in the cytoplasm of glioma cells. 

To further check the CTX-CNF1 biodistribution in the brain, we performed immunoblots of GL261-injected animals that had received the chimeric treatment (intraventricularly or intravenously). Using the anti-CNF1 antibody that recognizes the catalytic domain (previously tested on CTX-CNF1, as shown in Appendix A), we found that the presence of the chimeric protein was detectable only in the glioma mass, but not in the surrounding peritumoral tissue nor in the motor cortex (used as an internal control; Figure 3E). These data point out that our recombinant protein shows high selectivity for glioma cells, when systemically administered. It is worth noticing that the anti-CNF1 antibody selectively targets the catalytic domain, which is cleaved once the toxin has entered into the cells [4]. Although CTX-CNF1 is around 118 KDa, we found a band of 33 KDa from samples of tumoral mass that reflects the size of the catalytic domain. 

## 3. Discussion

Nowadays the main challenge for neuro-oncologists is to find effective approaches to counteract glioma growth and preserve the surrounding brain healthy tissue [4]. Progress in treating GB should require a specific and adequate delivery of the right drug to glioma cells, avoiding possible therapy side effects and deterioration of the peritumoral tissue [19]. 

In the last few years it has become clear that toxins’ peculiarities can be exploited for the treatment of cancer. As a matter of fact, toxins are extremely effective enzymes with great selectivity towards specific cellular substrates and, importantly, they are not subjected to drug resistance mechanisms [20]. Furthermore, they can be easily modified: their active core can be isolated and cloned to more efficiently penetrate into solid tumors, or they can be combined with carriers and antibodies to specifically enter cancer cells [4,5]. 

Here we presented CTX-CNF1, a recombinant molecule that we developed from the conjugation of two toxins. On one side it has CNF1, recently pointed out as an antineoplastic agent and as a neuronal function keeper [3,12,21,22]. On the other side there is CTX, known for passing the BBB and already largely employed in clinics as tumor paint and/or as drug vector in glioma clinical trials. Importantly, CTX also showed minimal cross-reactivity with healthy brain, making this protein particularly safe for its use on humans [7,10,11,23]. The reason of fusing these two molecules lies on the fact that native CNF1, despite its potentiality to treat gliomas, is incapable of selectively entering glioma cells and crossing the BBB. Thus, by conjugating CNF1 with CTX we created a new molecule that we used to treat glioma cell lines. Importantly, such cell lines express receptors for both these two toxins (i.e., Annexin A2 and MMP2 for CTX and Lu/BCAM for CNF1) [24,25,26]. CTX-CNF1 not only provokes senescence and cell death in several glioma cells (i.e., GL261, PDGF^+^ Trp53^−/−^, U87; Figure 1C–H), but above all penetrates the BBB, recognizing and targeting tumor cells with high specificity (Figure 3). As a consequence, CTX-CNF1 treatment resulted in effectively increasing glioma-bearing mouse survival, even when the administration was made through the caudal vein (Figure 2B,D). This result is particularly relevant because the major limit of cancer drugs developed for GB treatment is that they do not easily cross the BBB, failing to reach the target cells when systemically delivered [19]. Hence, creating a drug that is able to pass the BBB and that results very specific for binding GB cells is of particular value, because it might guarantee a more precise targeting of glioma cells with limited side effects. Another relevant point is that CTX-CNF1 was given in vivo and at a symptomatic stage of the disease, thus increasing the relevance of its possible translation to clinics. In order to make a good comparison between toxins, we performed the systemic administration (i.e., intravenous) of both CTX and CNF1 alone and we checked the survival rate of those mice. As expected, CNF1 alone was toxic and lethal for the 66% of glioma-bearing animals, failing to produce an enhanced survival for the remaining 33% (Figure 2F, [5]). On the other hand, the systemic delivery of CTX alone increased the survival rate of glioma-bearing animals (Figure 2H, *p* = 0.03). We also checked whether the enhanced survival produced by chimeric treatment was paralleled by a decreased in the proliferation rate of glioma cells. We found that, 48 h after chimeric administration, a significant reduction of Ki67 positive cells was visible in CTX-CNF1 treated animals but not in the CTX treated group (Figure 2I). No analysis on proliferation was made for CNF1 treated mice, because of the lack of impact on the survival rate of this therapeutic protocol (Figure 2F). Importantly, except for CNF1 intravenous administration, no toxic acute effects were observed in glioma-bearing animals treated with the other protocols (data not shown). 

Modeling gliomas is essential for the development of effective treatments. However, testing potential drugs on different models appears necessary, because each model presents only some characteristics of the disease and a model that fully recapitulates human GB is not established yet [27,28]. Hence, the high potentiality of our recombinant molecule has been confirmed by the fact that CTX-CNF1 was found to have similar effects on three different preclinical glioma models (i.e., GL261, U87 and PDGF^+^ TRP53^−/−^). 

After CTX-CNF1 in vivo delivery we reported an increased survival rate together with a decrease of glioma proliferation (Figure 2I). We hypothesized that this event might be due to the senescent process induced by the chimeric molecule on tumor cells (Figure 1D–F), that might recall the innate immune system to the GB-treated mass (Figure 2L,M). Indeed, several studies have suggested that CNF1-induced Rho GTPase activation might directly affect the functionality and pro-inflammatory potential of immune cells [29]. However, although we did not find any differences in the number of peritumoral Iba1-positive cells, we reported a significant increase of MAC2-positive cells in CTX-CNF1 treated mice (Figure 2). Interestingly, a higher expression of MAC2 in patients seem also to be linked to a better prognosis and to a better response to therapies [30]. 

In summary, taken altogether, our results provide evidence that CTX-CNF1 could represent an innovative and effective approach for GB treatment of high translational value. Of course, despite the promising results here presented, further studies still need to be done in order to validate the potentialities and the mechanisms of action of this novel chimeric conjugate.

## 4. Materials and Methods 

### 4.1. Cloning and Preparation of CTX-CNF1

The sequence encoding Chlorotoxin (36 amino acids, 108 nucleotides: AATGTGCATGCCGTGTTTCACCACCGATCACCAGATGGCCCGTAAATGCGATGATTGTTGCGGTGGTAAAGGTCGTGGTAAATGCTACGGTCCGCAGTGTCTGTGCCGTTG) was cloned as G-block (with BamH1-BglII restriction sides) in frame into the vector pGEX2TGL+2-CNF1 opened with BamH1 and dephosphorylated (corresponding to an N-terminal extension of CNF1). The correct cloning was verified by sequencing. 

GST-CTX-CNF1 or GST-CNF1 proteins were expressed in *Escherichia coli* BL21 cells transformed with pGEX plasmids carrying the respective genes. Bacteria were grown in Luria–Bertani (LB) medium at 37 °C and induced with 0.2 mM isopropyl-β-D-thiogalactopyranoside (IPTG) at an optical density of 0.6. The cells were harvested after 6 h, and proteins were purified by affinity chromatography with glutathione sepharose beads (Amersham Pharmacia Biotech). Loaded beads were washed two times in washing-buffer A (20 mM Tris/HCl pH 7.4, 10 mM NaCl, 5 mM MgCl2) and washing-buffer B (150 mM NaCl, 50 mM Tris/HCl, pH 7.5) at 4 °C. The proteins were eluted with 10 mM glutathione in 50 mM Tris/HCl pH 8.0. GST fusion proteins were lyophilized and stored frozen. Proteins were used as GST-fusion proteins because of higher stability.

### 4.2. Cell Cultures

GL261 and U87 cells were grown in complete Dulbecco’s modified Eagle’s medium (DMEM) containing 10% Newborn calf serum, 4.5 g/L glucose, 2 mM glutamine, 100 UI/mL penicillin and 100 mg/mL streptomycin at 37 °C in 5% CO2 with media changes three times per week [31,32]. Cells derived from the tumor mass of PDGF^+^ TRP53^−/−^ glioma-bearing mice were grown in DMEM containing 4.5 g/L glucose, 2 mM glutamine, 100 UI/mL penicillin, 100 mg/mL streptomycin, 10 ng/mL EGF, 20 ng 7 mL FGF, 1:50 B27 supplement at 37 °C in 5% CO2 with media changes two times per week. PDGF+ TRP53^−/−^ cells were a kind gift from Dr Claudio Giachino (Basilea, Switzerland; [16]).

### 4.3. MTT and Senescence-Associated Beta-Galactosidase (SA Beta-Gal) Assay

The MTT assay was performed on GL261 cells 48 h after CTX-CNF1 administration, following the protocol of ATCC bioproducts (Teddington, UK). The chimeric protein was given at different concentrations (1, 12, 25, 30, 40 and 50 nM) and assessed in 3 wells per condition. To determine cellular senescence, GL261 cells were plated in triplicate at low density (50% confluence) in 24-well plates; they were treated with CTX-CNF1 (25 nM) and incubated for 24, 48, 72 h or 6 days. The percentage of positively stained cells (senescence detection kit, Abcam catalog ab65351) was determined after counting 3 random fields using ImageJ software. Multi- and mononucleated GL261 cells were calculated after 48 h of 25 nM CTX-CNF1 administration.

### 4.4. Real-Time PCR

After 6 days of Vehicle/CTX-CNF1 (25 nM) treatment, quantitative real-time (RT) PCR reactions were performed on GL261 and U87 cells using the SYBR PCR Mastermix (Applied Biosystems) on a StepOnePlus Real-Time PCR System (Applied Biosystems).

### 4.5. Formation of PDGF^+^ TRP53^−/−^ Multicellular Spheroids

To determine the CTX-CNF1 effect on PDGF^+^ TRP53^−/−^ cells, PDGF^+^ TRP53^−/−^ cells were cultured and treated with the chimeric protein for 72 h or 6 days. At each time point, the number of total cells and tumorspheres spheroids was counted using ImageJ software.

### 4.6. Animals and Tumor Induction

Adult (age > postnatal day 60) C57BL/6J mice were used. All experimental procedures conformed to the European Communities Council Directive #86/609/EEC were approved by the Italian Ministry of Health (260/2016-PR, released on 11/03/2016). To induce glioma formation, C57BL/6 mice received a stereotaxically guided injection of 40,000 GL261 cells (20,000 cells/μL phosphate buffered saline solution; [33]) or 30.000 PDGF+ TRP53- cells (15.000 cells/μL phosphate buffered saline solution; [16]) into the visual cortex (2.5 mm lateral to the midline and in correspondence with lambda). 12 days after, mice were divided into four groups: the first group received CTX-CNF1 injection in the lateral ventricles (1 uL for each ventricles, 80 nM; coordinates: ±1 mm lateral and 0.8 mm posterior to bregma suture; “iv”), the second was infused intravenously with CTX-CNF1 (20 uL, 80 nM; “en”); the third and the fourth groups received phosphate buffered saline solution intraventricularly or intravenously respectively (i.e., control groups). Other cohorts of glioma-bearing animals received CTX or CNF1 alone through the caudal vein 12 days after tumor induction. Animal weight was checked daily and when the weight loss reached 30% of the baseline weight, animals were sacrificed. 

### 4.7. Western Blot

In order to avoid circadian effects, all animals were sacrificed during the same time interval each day (10–12 h; light phase). After decapitation, brains were rapidly removed and various brain areas were collected. All the tissues were collected 48 days after CTX-CNF1 administration, dissected and frozen on dry ice. Proteins were extracted as reported in [2]. Briefly, protein extracts were separated by electrophoresis and blotted; filters were blocked and incubated overnight at 4 °C with anti-CNF1 catalytic domain primary antibodies (1:1000, ThermoFisher, Waltham, MA, USA) and probed with anti-tubulin antibody (1:40,000, Cell Signaling, Danvers, MA, USA) as an internal standard for protein quantification.

### 4.8. Immunohistochemistry

To quantify the localization of CTX-CNF1 we performed a staining for CNF1 (1:500, ThermoFisher). The PDGF^+^ TRP53^−/−^ cell line was GFP^+^ [16]. Hoechst staining was also performed (1:500, Sigma, St. Louis, MI, USA). To evaluate number of Ki67+, Iba1+ and Mac2+ cells, mice were deeply anesthetized and perfused with 4% paraformaldehyde 48 h after CTX-CNF1 administration. Coronal brain sections through the occipital cortex (45 μm thick) were cut with a freezing microtome. Brain serial sections were stained with Iba1 (1:500, Wako, Tokyo, Japan) and Mac2 (1:400, Fitzgerald, Birmingham, UK) to assess the microglia/macrophages number; cell counting was carried out using ImageJ software. For the evaluation of glioma proliferation, brain serial sections were stained with Ki67, a common used marker of proliferation (1:400, Abcam, Cambridge, UK). After incubation with Iba1, Mac2 or Ki67 antibodies, slices were incubated with fluorophore-conjugated secondary antibodies (Jackson Immunoresearch) and with Hoechst dye (1:500, Sigma) for nuclei visualization. 

### 4.9. Image Acquisition and Analysis

Fluorescent images were acquired using a Zeiss Axio Oberver microscope equipped with Zeiss AzioCam MRm camera (Carl Zeiss MicroImaging GmbH, Jena, Germany) and using a Zeiss LSM 900 with Airyscan 2. For the quantification of the density of KI67+ cells, the whole tumor area was acquired for each coronal section. Images were processed with ImageJ software (National Institute of Health) and the density of proliferating cells was expressed as the fraction of area occupied by KI67+ cells with respect to total glioma area [34]. For comparison between treatments, at least 3 animals per group were analyzed. 

### 4.10. Statistical Analysis

GraphPad 8 software was used for statistical analysis. The differences between the two groups were evaluated with *T*-test, whereas differences between three or more groups were assessed with one-way ANOVA. Survival analysis was performed using Kaplan–Meier (LogRank) statistics. 

## Figures and Tables

**Figure 1 toxins-13-00194-f001:**
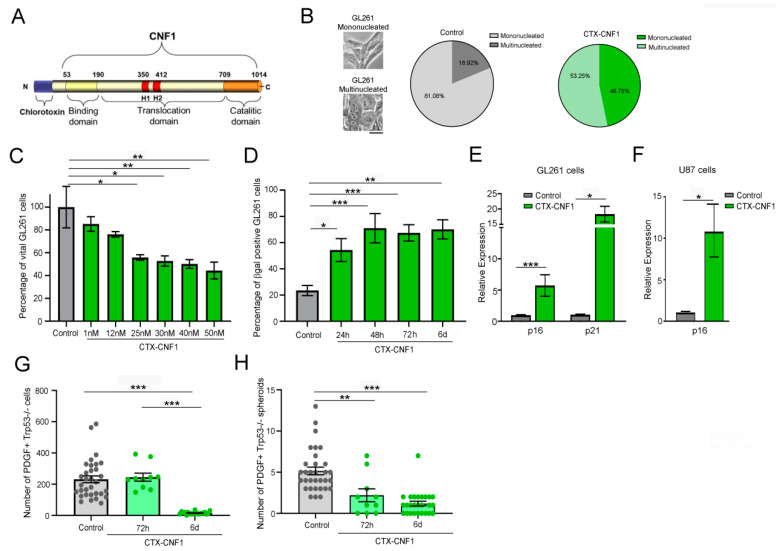
CNF1 maintains its activity on glioma cells even when inserted in the CTX-CNF1 recombinant molecule. (**A**) Structure of CTX-CNF1. Chlorotoxin (CTX) is located at the N-terminus of the recombinant molecule and is followed by CNF1, formed by its three domains (binding, translocation and catalytic). (**B**) Percentage of multi- and mononucleated GL261 cells in control (grey, PBS) and after 48 h of CTX-CNF1 administration (green, 25 nM). Inset, representative images of the two conditions, scale bar = 4μm. (**C**) Percentage of vital GL261 cells in control (grey, PBS) and in 48 h treatment of CTX-CNF1 (green) at different concentrations (i.e., 1, 12, 25, 30, 40 and 50 nM). Data represent means ± SEM, one way ANOVA *p* < 0.001. (**D**) Percentage of betagalactosidase positive GL261 cells in control (grey, PBS) and after CTX-CNF1 administration (green, 25 nM) at different time points (i.e., 24 h, 48 h, 72 h, 6 d). Data represent means ± SEM, One Way ANOVA *p* < 0.001. (**E**) Quantitative RT-PCRs showing the relative expression of the senescence markers p21 (Cdkn1a, cyclin-dependent kinase inhibitor (1) and p16 (Cdkn2a, cyclin-dependent kinase inhibitor 2) in GL261 cells (vehicle, grey; CTX-CNF1, green). Data represent means ± SEM, *t* test (p16, *p* = 0.0001; p21, *p* = 0.0164). (**F**) Increased expression of p16 (Cdkn2a, cyclin-dependent kinase inhibitor 2) in U87 cells treated with CTX-CNF1 (green) with respect to vehicle (grey). Data represent means ± SEM, *t* test (*p* = 0.013). Total cells (**G**) and spheroids (**H**) number of PDGF+ TRP53−/− cells in control (grey, PBS) and after 25 nM of CTX-CNF1 (green) at 72 h and 6 d. Data represent means ± SEM, One Way ANOVA *p* < 0.001. * *p* < 0.05; ** *p* < 0.01; *** *p* < 0.001.

**Figure 2 toxins-13-00194-f002:**
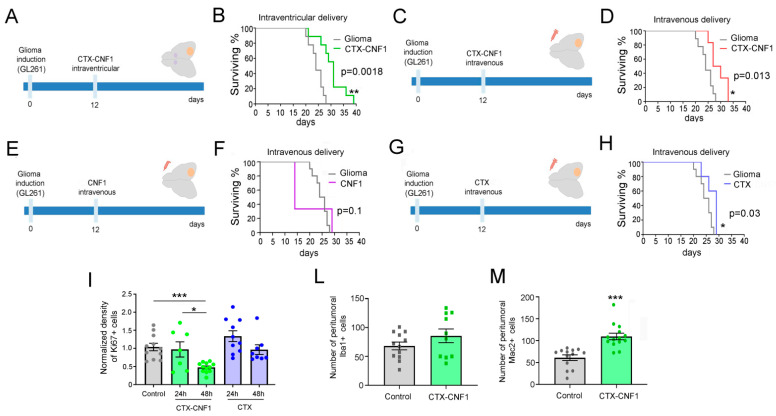
Increased survival of CTX-CNF1 glioma-bearing mice. (**A**,**C**,**E**,**G**) Experimental protocols. GL261 cells were injected in visual cortex and, 12 days after glioma induction, different treatments were administered. (**A**,**C**) Glioma-bearing mice were treated intraventricularly (**A**,**B**) or intravenously (**C**,**D**) with CTX-CNF1 (80 nM) or vehicle solution (PBS). (**B**,**D**) Survival rate of untreated and treated glioma-bearing mice. Note the increased surviving percentage of both CTX-CNF1 treated glioma-bearing mice (**B**, *p* = 0.0018; **D**, *p* = 0.013). * *p* < 0.05; ** *p* < 0.01. (**F**,**H**) Survival rate of untreated and intravenously treated (**F,** with CNF1; **H**, with CTX) glioma-bearing animals. We did not find any significant increase in survival after CNF1 (**F**, *p* = 0.1); conversely, CTX treatment was able to increase survival rate of glioma-bearing animals (**H**, *p* = 0.03). (**I**) Normalized fraction of tumor area occupied by KI67+ cells in control and in animals intravenously treated with CTX-CNF1 or CTX alone, 24 h and 48 h after the toxin administration (one way ANOVA *p* < 0.001). Data represent mean ± SEM. *** *p* < 0.001. (**L**,**M**) Number of peritumoral Iba1+ (**L**, *p* = 0.19) and Mac2+ (**M**, *p* < 0.001) cells in mice injected with GL261 cells into the visual cortex and systemically treated with CTX-CNF1 with respect to controls. Data represent mean ± SEM. *** *p* < 0.001.

**Figure 3 toxins-13-00194-f003:**
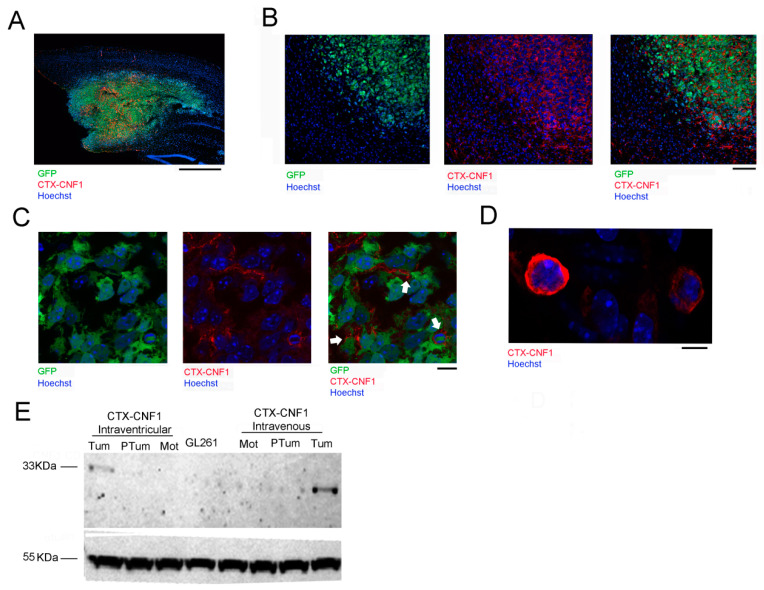
CTX-CNF1 specificity of action on glioma cells. Representative image (**A**) and magnification (**B**) of a triple staining for chimeric protein (red), glioma PDGF+ TRP53^−/−^ GFP^+^ cells (green) and cellular nuclei (blue). Note the high selectivity of CTX-CNF1 for recognizing and entering into GB mass. CTX-CNF1 was intravenously delivered and the pictures were taken 24 h after chimeric administration. (**A**) Scale bar = 1000 μm. (**B**) Scale bar = 50 μm. (**C**) In vivo image taken of PDGF^+^ TRP53^−/−^ GFP^+^ cells 24 h after CTX-CNF1 intravenous delivery. Scale bar = 5 μm (**D**) In vivo image taken of GL261 cells 6 days after CTX-CNF1 intravenous delivery. Scale bar = 10 μm. (**E**) Representative western blot: 12 days after GL261 injection, CTX-CNF1 administration were performed (left, intraventricular; right, intravenous). 48 h after the treatment, the chimeric protein is clearly detectable only in the tumoral mass, even when its administration was made through the caudal vein. Tum = tumor; PTum = peritumoral; Mot = motor area; CD = catalytic domain; GL261 = GL261 cells lysate untreated.

## Data Availability

No new data were created or analyzed in this study. Data sharing is not applicable to this article.

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
