# Peer review of "CTX-CNF1 Recombinant Protein Selectively Targets Glioma Cells In Vivo"

_toxins, 2021, doi:10.3390/toxins13030194_

Round 1
Reviewer 1 Report
The manuscript has been improved in thjis neu version. I have no further comments.
Author Response
We would like to thank the Reviewer for his/her careful reading of our manuscript.
Reviewer 2 Report
The novel medical use of CNF1 is of widespread interest, and in this study the authors design and evaluate a potentially useful therapeutic agent for the treatment of gliomas. The work is well carried out, but in places needs a clearer explanation. In particular the multipart figures 2 and 3 should have some more annotation and clearer figure legends.
I list a number of suggestions for improving the English.
Line 124 The rationale for using Iba1 and Mac2 as markers needs to be introduced.
Figure 2 This figure legend is confusing. The statement that "GL261 cells were injected... after glioma induction" should be "recombinant proteins ..." The glioma induction was by using GL261 cells. The ACEG protocols diagram needs more explanation.
The figure 3 legend needs to note that GFP is a marker for PDGF+/TRP53-/- cells. It is unclear where the molecular weight markers are in Figure 3E. There should be lines to mark the intended molecular weights and numbers given instead of CNF1 CD.
Line 35 "In the last few years our group has studied" would be better
Line 57 "Regardless of"
Line 61 "53.25%" I doubt that this figure is accurate to 4 places
line 65 should be "after only"
Line 72 do they mean "rich in"?
Line 73 "The number of.."
Line 89 "relative expression" - plural not usual
Line 146 "versus the CNF1 domain"
Line 156 "but not ... nor in ..." is better
Line 158 Either "the glioma cell" or "glioma cells"
Line 161 "Although CTX-CNF1 is ....from samples of tumoral mass that reflects the size of the catalytic domain"
Line 180 "In the last few years"
Line 185 "to specifically enter cancer cells"
Line 194 "incapable" instead of "unable"
Line 195 "crossing the BBB"
Line 201 "resulted in effectively increasing"
Line 201 "mouse survival"
Line 206 "specifically"
Line 207 "with limited side effects"
Line 212 "alone was toxic..."
Line 239 "evidence" it is never used in the plural
Line 240 "represent"
Line 287 "the CTX-CNF1"
Line 304 "Animal weight"
Line 317 "The PDGF...."
Line 322 "cell counting"
Line 325 "After treatment with Iba1..."
Author Response
We thank the Reviewer very much for the careful reading of our manuscript and for his/her suggestions, that we really appreciate.
We have now made all the requested changes (in red in the text).

This manuscript is a resubmission of an earlier submission. The following is a list of the peer review reports and author responses from that submission.
Round 1
Reviewer 1 Report
In this manuscript, authors generate a chimeric protein formed by the Cytotoxic Necrotizing Factor 1 (CNF1) and Chlorotoxin (CTX). The manuscript represents a preliminary study showing the cytotoxic effect of the chimeric protein on glioma cell lines and the survival time in “in vivo” experiments using mice with experimental glioma (by injecting same glioma cell lines into mouse visual cortex)
These efforts are welcome since gliomas are very aggressive brain cancers and the results are promising, however additional data to support the results is strongly required.
Comments:
- Authors claim CTX-CNF1 is a new chimeric protein created by fusion of CTX with CNF1. Being a new molecule it should be well characterized and complete data provided:
- how was CTX-CNF1 DNA produced and cloned?
- how was the recombinant protein produced?
- how was it purified?
- how pure was the chimeric protein preparation, were any contaminants present?
- CTX-CNF1 has a toxic effect on glioma cell lines, but which is its toxic activity compared to the wild type CNF1?
- The CNF1 is a rather big protein, compared to CTX, thus demonstrating the pass of CTX-NCF1 through the blood brain barrier is encouraging. The antibody used identifies the catalytic domain of CNF1, and western blot in figure 3 shows a positive band of 33 kDa. However, a 110 kDa band should be expected, otherwise it should be explained how the catalytic domain was segregated from the CTX-NCF1 fusion protein (figure 3C). Was an additional fusion protein CTX-CD-NCF1 (with just the catalytic domain of CNF1) administered to mice in parallel studies? In any case, it was not the full chimeric protein detected in the western blot (figure 3C).
- Tumors were induced by stereotaxically guided injections of the glioma cell line. In this series of experiments, a control group of animals with similar stereotaxically guided injections but without glioma cells, should be added. Stereotaxic injections can produce lesions which may interfere with the blood brain barrier permeabilization studies.
- Figure 2B and D. Numbers in the x-axis (days) must be revised.
- Images in figures 2G are too small and of low resolution.
- Figure 3. Immunofluorescence staining shows the presence of the chimeric protein (using the anti-CNF1 catalytic domain antibodies) in the tumoral mass (after injection of PDGF+ TRP53- cell line GFP+):
- Please confirm the brain region shown in figure 3A.
- It is assumed the CTX-CNF1 chimeric protein is internalized into glioma cells. Accordingly the chimeric protein and the GFP+ cells should colocalize. However, although figure 3B show the presence of both CTX-CNF1 and GFP+ cells in the same region, a real colocalization (green and red do not coincide, and the yellow mark is absent in the ) is not shown. An image with higher resolution would help.
Reviewer 2 Report
In their manuscript „CTX-CNF1 recombinant protein selectively targets glioma cells in vivo” the group investigate a novel strategy for glioma treatment. Aiming to increase tumor cell specificity as well as blood brain barrier (BBB) penetration of promising treatment options like the cytotoxic necrotizing factor (CNF1), they developed a chimeric protein conjugating CNF1 to chlorotoxin (CTX). Overall, the subject is very interesting. There are, however, a few points I would like to raise as the paper clearly could do better with some additional work.
- Paper requires another round of proof-reading (wrong grammar, missing spaces between numbers and their units, units should be checked carefully).
- Introduction: Introduction needs expanding to explain the context. Possibly a figure graphically summarizing the group’s previous findings on CTX and CNF1 in glioblastoma cells would be very helpful for the reader to understand the context.
- Introduction - page 1, line 27: Predominantly called Glioblastoma now days, Glioblastoma multiforme is obsolete (but still used in the literature).
- Page 2, line 65: Wrong reference? à should be [28] Giachino et al., Cancer Cell, 2015, or?
- Figure legends need to be expanded to have all essential information present. It is not clear how often experiments were performed (including technical as well as biological replicates) and what type of readout was used. Materials and methods section is also missing essential information and needs to be revised.
- In general – with the aim to proof that the CTX-CNF1 conjugate can cross the BBB and specifically targets glioma cells:
- The correct control for all experiments would not be PBS only, but in addition also CNF1 only, CTX only and CNF1 conjugated to a scrambled peptide to verify the specificity of this approach.
- It is required to evaluate the cellular targets of each component – which receptor is required for glioma specific binding and are these expressed in the model used in the current paper – if this is unknown, it should be discussed at least!
- Figure 1:
- Orientation of the figures is confusing.
- B:
- The IC50 is specific for each time point of analysis and should be calculated of a series of dose-response data whereas 3 different concentrations are, from my point of view, too less to do it properly.
- A line chart is inappropriate for this type of data. A column chart should be used instead.
- C:
- Representative pictures of mono- and multinucleated cells as well as adding the percentage of the respective fractions in the pie chart would improve the quality of the figure.
- Unclear at which time point this analysis has been done and how it has been done – should be added to the figure legend and the material and methods section.
- D: β-Galactosidase staining on its own is not a definitive marker for senescence induction. Further markers should be evaluated in this context.
- Why? à see C+D:
- The fraction of multinucleated cells corraltes with the fraction of cells that are β-Gal positive.
- Question: Are multinucleated cells stressed and therefore, show signs of senescence without being truly senescent?
- E+F: wrong y-axis label à should be PDGF+ instead of PDFD+.
- Results – page 3, line 85-110:
- A short introduction into the following section would improve readability.
- Blood brain barrier abbreviation was already introduced in the introduction section and should be used instead.
- Units should be checked carefully.
- Check grammar carefully.
- Results – page 4, line 96-101:
- What´s the rationale of this experiment? Moving the appropriate information from the discussion to the result section would make this section easier to follow.
- If aiming to assess the infiltration of immune cells that might be responsible for clearing senescent cells, the quality of the paper would be much improved by including additional markers for other subsets. Currently, the figure/data, however, from my point of view does/do not contribute much to the whole story.
- Personally, I wonder if the paper would not read better, if figure 2 E,F and G were excluded and a more detailed characterization of the model (i.e. tumor burden, staining for proliferation markers like Ki67, staining for senescence markers, changes in invasiveness of tumor cells upon treatment) would be added instead.
- Figure 2:
- B+D: add statistics to the figure – so far only mentioned in result section.
- G: The quality of the figure would be much improved when these microscopic pictures would be enlarged, and a negative control must be included.
- Figure 3:
- A+B:
- Unclear which cohort has been used for this immunofluorescent staining – intraventricular or intravenous administration?
- Clearly state which cells have been used in the model – only mentioned in the text, but not in the figure legend.
- If B represent a magnification of figure A, it would be an improvement to highlight the respective region B has been imaged from.
- C:
- Representative of how many blots?
- Loading control shows unequal loading of samples – could be easily improved.
- According to its data sheet, α-Tubulin has a size of 55 kDa but in the blot it was found at around 50 kDa?
- Clearly state in the figure legend which cells have been used in the model and which type of administration was used.
- In general: Toxicity profile – blood, other organs? à CTX and CNF-1 on their own do not induce any apparent signs of toxicity but it should be proven for the conjugate, too.
- One of the main reasons glioblastoma is a difficult to treat disease is its invasive growth pattern thereby affecting the whole brain. Consequently, maximal safe surgical resection aims to alleviate symptoms but does not cure the patient. This characteristic feature is often not or only partially recapitulated by cell line-derived tumors in vivo. GL261-derived tumors are characterized by an only very modest invasion compared to human-derived tumors. PDGF+TP53-/- cell culture as well as PDGF+ TP53-/- -derived tumors, in general, should be characterized particularly regarding invasiveness and other typical growth characteristics associated with glioblastomas.
- A+B:
Reviewer 3 Report
Title: CTX-CNF1 recombinant protein selectively targets glioma cells in vivo
Article Type: Original Research
Submitted to: Toxins
Opinion: Accept with minor revisions/major revisions
Summary: This original research article advances a novel approach for glioma treatment through the development and characterization of a chimeric protein of CNF1 and CTX. Previous research showed that CNF1 provided reduces tumor mass, but only through direct injection into the brain. This work fuses the protein CTX, known to cross the blood-brain barrier, to CNF1 and both deliver to the brain as well as efficacy is established.
This work is well-designed, well-organized, and with excellent figures. It does, however, have a tendency to overstate some findings and more detail is needed in some areas. While the organization of the paper is excellent with a clear structure, the paper has many grammatical errors that should be corrected prior to publications. Lastly, the authors cite their own work too often. Over ¼ of the citations are self-citations, and it is especially noticeable in some areas as in multiple self-citations for the culturing of a common cell line.
Most importantly, the paper states that CNF1 cannot cross the blood-brain barrier. Has this experiment ever been explicitly performed? If it hasn’t, that experiment must be performed. Otherwise, there is no way to tell that CNF1 did not inherently contain some amino acid sequence that allows the crossing of the blood-brain barrier, not from the CTX sequence.
Minor revisions
- Line 47: The authors claim CTX-CNF1 “maintained unaltered the properties of the two components.” This is not shown and is also unlikely. CTX alone most likely crosses the membrane more easily than CTX-CNF1. This statement would require a comparative quantification of CTX administered alone. Likewise, a direct comparison to injected CNF1 would be needed to quantify cytotoxicity. The paper does show that CTX-CNF1 contains measure of both components, not that it’s unaltered or undiminished.
- Order of Figure 1 is off (ABECDF?)
- Figure 1B, this is not necessary to perform for publication, but it would be nice to know at what concentration cell death saturates.
- Line 56, it’s “statistically” not “strongly”. The paper often writes “strong” differences, which is not correct usage. The differences are “statistically significant”.
- The full DNA sequence needs to be provided, with a clear explanation on where the fusion site is.
- The sentence in Line 43 is unclear “Indeed….”
- Line 183: Cell culture conditions should be described, instead of a multiple self-citations (over ¼ of citations are self)
- Line 70, “CNF1 kept its functionality” is incorrect, there was no comparison with injected CNF1. You can state “kept some functionality”
- Line 196-197, were GL261 cells also counted with ImageJ? More detailed description of how cells were counted with ImageJ is needed.
- Line 86, the first sentence of the paragraph should describe how the glioma bearing mice were created, as opposed to jumping straight to “12 days…”
- Line 87, “randomly” should be removed. Also, for “group of animals” how many?
- The method section should provide the DNA sequence of CTX-CNF1, the culturing conditions in E. coli, and protein purification steps.
- Line 95, “highlight managed to cross” was not shown in this figure, only efficacy, not how the chimera was effective. In Figure 3, you show that the protein managed to cross.
- Line 96, you should discuss why having more microglial cells is important
- Line 100. “strong” increase means significant.
- No discussion of Figure2G in text
- Line 133: “innovative” approaches are not needed, “effective” approaches are.
- Line 153 (Figure 2 does not show crossing the barrier, only Figure 3)